# Primary Screening of Microorganisms against *Meloidogyne hapla* (Chitwood, 1949) under the Conditions of Laboratory and Vegetative Tests on Tomato

**DOI:** 10.3390/plants12183323

**Published:** 2023-09-20

**Authors:** Svetlana N. Nekoval, Arina K. Churikova, Maxim N. Chernyakovich, Mikhail V. Pridannikov

**Affiliations:** 1Federal Research Center of Biological Plant Protection, p/o 39, 350039 Krasnodar, Russia; arina.churikova98@mail.ru (A.K.C.); chernyakovich96@mail.ru (M.N.C.); 2Center of Parasitology “A.N. Severtsov Institute of Ecology and Evolution of the Russian Academy of Sciences”, 33 Leninsky Prospekt, 119071 Moscow, Russia; mikhail.pridannikov@yahoo.com

**Keywords:** microorganisms, root-knot nematodes, *Meloidogyne hapla*, nematicidal activity, galls, tomato, nematicides

## Abstract

Highly adapted obligate endoparasites of the root system, root-knot nematodes (*Meloidogyne* spp.), cause great damage to agricultural crops. Our research is aimed at the assessment of nematicidal activity and effectiveness of antagonist fungal and bacterial strains against the most common type of root-knot nematode in the south of Russia. By means of molecular genetic identification, it was found that in the south of Russia, the species *Meloidogyne hapla* Chitwood, 1949 and *Meloidogyne incognita* (Kofoid and White, 1919) Chitwood, 1949 cause galls on the roots of open-ground and greenhouse tomato. Screening of microbial agents against second-stage juvenile (J2) *M. hapla* was carried out in the laboratory. At the end of the experiment, two liquid fungal cultures of *Paecilomyces lilacinus* BK-6 and *Metarhizium anisopliae* BK-2 were isolated, the nematicidal activity of which reached 100.0 and 70.2%, and exceeded the values of the biological standard (Nemotafagin-Mikopro) by 38.4% and 8.8%. The highest biological efficacy was noted in the liquid cultures of *P. lilacinus* BK-6, *M. anisopliae* BK-2, and *Arthrobotrys conoides* BK-8 when introduced into the soil before planting tomato. The number of formed galls on the roots was lower in comparison with the control by 81.0%, 75.5%, and 74.4%.

## 1. Introduction

Root-knot (gall) nematodes (*Meloidogyne* spp.; RKNs) are highly adapted obligate endoparasites of the root system that cause a disease known as meloidoginosis [1]. They infect up to 2000 plant species [2].

According to world scientists [2,3,4,5,6], on vegetable crops with meloidoginosis, the following species are found in 95 cases: *Meloidogyne incognita* (Kofoid & White, 1919) Chitwood, 1949 (southern root-knot nematode), *M. hapla* Chitwood, 1949 (northern root-knot nematode), *M. javanica* (Treub, 1885) Chitwood, 1949 (Javanese root-knot nematode), *M. arenaria* (Neal, 1889) Chitwood, 1949 (peanut root-knot nematode), *M. chitwoodi* Golden, O’Bannon, Santo & Finley, 1980 (Columbia root-knot nematode), and *M. fallax* Karssen, 1996 (false Colombia root-knot nematode). They cause significant damage to crops, with yield losses exceeding 30% [7,8].

The most widely used method of dealing with RKNs is chemical. The first preparations against phytoparasitic nematodes appeared at the beginning of the twentieth century [9]. These were broad-spectrum chemical fumigants: nematicides, fungicides, insecticides, and herbicides. It was noted that chemicals in the form of gases penetrating the soil kill not only target objects, but also nematodes. A negative result from the use of fumigants was phytotoxicity. Therefore, it is advisable to use them only before planting crops [10,11].

Between 1957 and 1992 non-fumigants were discovered: carbamates (aldicarb, carbofuran, and oxamyl) and organophosphorus compounds (phensulfothion, ethotrop, fenamiphos, terbufos, cadusafos, and fostiazate). These compounds are characterized by relatively low phytotoxicity and selectivity towards harmful organisms. They are considered suitable for use during planting and after germination [12,13].

In modern crop production, both fumigants and non-fumigants continue to be used against meloidoginosis. According to Desaeger et al. (2020), and Oka (2020), new non-fumigants are nematicides, which include the chemical compounds fluopyram, fluensulfone, and fluazaindolizine. Fluopyram, initially employed as a fungicide, belongs to the benzamide class and functions as a succinate dehydrogenase inhibitor. Its nematicidal properties were discovered subsequently [12,14]. Most studies using fluopyram have been against *M. incognita*. The use of fluopyram under in vitro conditions and on greenhouse tomato crops was highly effective [15,16,17,18]. In Europe, preparations based on it are Verango, Velum, Indemnify [15,19], registered by Bayer CropScience. Fluensulfone, from the class of 1,3-thiazoles, is part of the Nimitz preparation, manufactured by Adama. It was registered in 2014 in the USA. The Salibro preparation is currently being registered in Canada, manufactured by DuPont/Corteva, of which the active substance is fluazaindolizine from the class of imidazopyridines [12,14].

In the Russian Federation, preparations based on carbamates, with oxamyl as the active substance, are registered and included in the *Reference Book of Pesticides and Agrochemicals Permitted for Use on the Territory of the Russian Federation 2023*. They are Vydat 5G, manufactured by DuPont Science and Technology LLC (Moscow, Russia), and Palitsa, manufactured by Agrokhim-XXI LLC (Stavropol Krai, Russia) [20].

However, according to the World Health Organization (WHO), the chemical compound oxamyl is categorized as Class I (extremely hazardous), fluopyram as Class II (moderately hazardous), and fluensulfone as Class III (slightly hazardous) in hazard classifications [21]. Despite their high efficiency, they adversely affect human health. For example, fluopyram and fluensulfone have a high (Class III) threshold of toxicological concern (TTC). They have mutagenic and carcinogenic effects on warm-blooded animals, including humans. They often lead to the development of oncological diseases [22,23].

Recognizing that the utilization of chemical plant protection products entails land degradation, soil, water, and air pollution, biodiversity reduction, as well as a negative impact on climate and human health, a process of transition to alternative methods of food production was launched [24]. A lot of scientific research around the world is aimed at finding new, alternative chemical methods of protecting plants against RKNs.

One of the environmentally safe approaches to control *Meloidogyne* species is the use of microorganisms that differ in their mechanisms of action: predatory fungi (*Arthrobotrys*, *Dactylella*, *Dactylellina*, etc.), endoparasitic fungi (*Trichoderma*, *Metarhizium*, *Drechmeria*, etc.), and fungi that is toxin-forming and parasitic on eggs (*Pochonia*, *Paecilomyces (= Purpureocillium)*, *Lecanicillium*, etc.) [25]. Also, scientists are actively working on the study of antagonist bacteria (*Bacillus*, *Pseudomonas*, *Pasteuria*, etc.), which protect plants from diseases, including meloidoginosis, and stimulate an active increase in plant biomass [26,27,28].

Earlier we carried out a search and analysis of the published data aimed at studying the diversity of nematicidal fungi and their mechanisms of action against RKNs. Our findings revealed that approximately 50 biological nematicides based on various types of fungi (*Arthrobotrys* spp., *Trichoderma* spp., *Purpureocillium lilacinum*, etc.) have been approved for use in countries and regions including France, South and North America, Africa, Europe, China, India, and others [29].

There is only one registered bionematicide on the Russian market (Nematofagin-Mikopro (*Arthrobotrys oligospora* F-1303)), and the problem of meloidogynosis remains relevant [30]. In this regard, it is necessary to search and study bioagents to create new, effective preparations against root-knot nematodes.

The aim of our research is to determine the species composition of root-knot nematodes, to assess the nematicidal activity and the effectiveness of fungal and bacterial antagonist strains to control the most common type of root-knot nematode in southern Russia.

Through the utilization of molecular genetic techniques, our study aimed to establish the *Meloidogyne* species composition affecting both open-ground and greenhouse tomato crops in the southern region of Russia. Additionally, we conducted screening experiments of fungi and antagonist bacteria in both in vitro and in vivo settings to assess their control potential over the most common type of root-knot nematodes.

## 2. Results

### 2.1. Identification of Root-Knot Nematode Species

Phytosanitary monitoring was carried out on various farms in the Rostov region (North-Western agro-climatic zone), Krasnodar Krai (Central and Anapo-Taman agro-climatic zones), the Volgograd region, and the Republic of Dagestan. Five populations of root-knot nematodes (*Meloidogyne* spp.) were isolated and maintained in vegetation pots.

It was found that root-knot nematodes selected from the farms of the Rostov region (M_hapla_1), Krasnodar Krai (M_hapla_2 and M_hapla_3) and Volgograd region (M_hapla_4) belonged to the species *Meloidogyne hapla* Chitwood, 1949 (Table 1), while those from the Republic of Dagestan (M_incognita_1) belonged to *Meloidogyne incognita* (Kofoid and White, 1919) (Table 2).

### 2.2. Effect of Liquid Cultures of Microbial Strains on Second-Stage Juveniles (J2) of M. hapla

As a result of the experiment, the effectiveness of the application of seven liquid cultures of microbial strains against J2 *M. hapla* juveniles was assessed in vitro (Figure 1).

Twenty-four hours after the start of the experiment, we observed a decrease in the mobility of juveniles and mortality in all experimental options, except for the control. When *Paecilomyces lilacinus* BK-6 and *Metarhizium anisopliae* BK-2 were introduced, the mortality of juvenile nematodes was higher by 45.0% and 5.0% compared to the biological standard (Nematofagin-Mikopro (*Arthrobotrys oligospora* F-1303)). In other options, nematicidal activity was lower.

Forty-eight hours after the start of the experiment, an increase in the mortality of individuals was observed in all experimental options (Figure 2). The number of immobile J2 did not exceed the values in comparison with Nematofagin-Mikopro, except for the options with *M. anisopliae* BK-2 and *P. lilacinus* BK-6. In the first case, mortality was observed on a par with the biological standard, and in the second case, the indicators were higher by 49.1%, which amounted to 98.3% of the immobilized J2. In the option with *A. conoides* BK-8, the mortality of juveniles was only 1.7% lower than in the biological standard.

At the end of the experiment (after 72 h), two strains of *P. lilacinus* BK-6 and *M. anisopliae* BK-2 were isolated, which were effective against the total number of J2. The nematicidal activity exceeded the biological standard by 38.4% and 8.8% (mortality: 100.0% and 70.2%), respectively.

### 2.3. Effect of Liquid Cultures of Microbial Strains on the Presence of Galls on Tomato Plants

The determination of the biological efficacy of seven microorganisms and the biological standard against the background of artificial soil infestation with the northern root-knot nematode (*M. hapla*) was carried out in 5 L vegetative pots on a susceptible mutant tomato line (Mo 463) (Table 3).

In the untreated control group, all the studied tomato plants were affected by the northern root-knot nematode, and the number of galls on the roots reached an average of 98.3 galls per plant (Figure 3). Soil treatment in growing pots with liquid cultures of microorganisms before planting tomato crops in all experimental options led to a different degree of reduction in the number of galls per plant. Biological efficiency ranged from 36.8% to 82.0%. The best result was noted with the introduction of *P. lilacinus* BK-6—81.2%, which is close to the biological standard—82.0%. We also highlighted options with *M. anisopliae* BK-2 and *A. conoides* BK-8. Their biological efficiency was lower than in the option with the introduction of Nematofagin-Mikopro, while reaching 75.7% and 74.5%, respectively.

## 3. Discussion

Our research on the study of the species composition of RKNs affecting both open-ground and greenhouse tomato crops allows us to conclude that in the south of Russia, meloidoginosis is caused by nematodes of the species *M. hapla* and *M. incognita*. Over most of its territory, galls on tomato roots form *M. hapla.* We consider this information to be unique, since the determination of the species composition of the genus *Meloidogyne* in southern Russia has not been previously carried out using molecular genetic methods.

We believe that the development and registration of the preparations against phytoparasitic nematodes based on living microorganisms and their metabolites, and the introduction of biological nematicides in the technology of growing vegetable crops, are important areas in the greening of agriculture.

In our studies, we assessed the effect of liquid cultures of strains of antagonist microorganisms on the northern root-knot nematode (*M. hapla*): *Metarhizium anisopliae* BK-2; *Arthrobotrys conoides* BK-8; *Arthrobotrys oligospora* BK-8/1; *Paecilomyces lilacinus* BK-6; *Trichoderma viride* G-2; *Bacillus thuringiensis* BK-10; and *Streptomyces avermitilis* BK-3. The work was carried out under laboratory conditions and under the conditions of a vegetative test on plants of the susceptible mutant tomato line Mo 463.

Many scientists have studied nematophagous fungi and antagonist bacteria exhibiting high nematicidal activity as agents of biological control against phytoparasitic nematodes [31].

It is known that *P. lilacinus* strains are the most widely used in the world as a biological control against root-knot and cyst-forming nematodes [32]. Thus, when using strains of the studied fungus, it was noted that they actively colonize the organic substrate and are considered an effective parasite of nematode eggs, which successfully control various types of root-knot nematodes on tomatoes, eggplants, potatoes, and other vegetable crops [33,34,35].

Our studies have proved the effectiveness of fungi *P. lilacinus* BK-6, *M. anisopliae* BK-2, and *A. conoides* BK-8. It has been found that the use of strains BK-6, BK-8, and BK-2 before planting the culture effectively reduces the formation of galls on a nematode-susceptible tomato plant. In vegetation experiments by German scientists, it was found that a single soil treatment with *Paecilomyces lilacinus* strain 251 (PL251) fungus before planting tomato reduced root damage by *M. incognita* by 66% [36]. Scientists from Egypt showed that before planting and after tomato infection with *M. incognita*, treatment with *Purpureocillium lilacinum* AUMC 10149 (= *Paecilomyces lilacinus*) reduced the nematode population, the number of galls, and the mass of eggs in plant roots [37]. The effectiveness of the predator fungus *A. conoides* against root-knot nematodes on tomato plants has been poorly studied under in vitro and in vivo conditions, which indicates the relevance of the research in this area.

Most studies on the nematicidal activity of the fungus *M. anisopliae* have been carried out against the species *M. incognita*. In an in vitro experiment, Turkish scientists obtained positive results using two isolates of *M. anisopliae*, where the effectiveness of the fungus against *M. javanica* and *M. incognita* after 24 h at a concentration of 108 CFU/mL was 98.5% and 97.1%. One of the isolates showed 100% mortality of juvenile nematodes after 48 hours, the other after 72 hours [38]. Scientists from India showed the activity of the fungus *M. anisopliae* against *M. incognita*, where 97.7% parasitism on J2 juveniles was noted. In addition, the studied isolate reduced the development of meloidoginosis in vivo by 82.3% [39]. We found that the effectiveness of *M. anisopliae* against the species *M. hapla* under laboratory conditions was 48.3–70.2%, and under the conditions of a vegetative test was 75.7%. These results show that the studied entomopathogenic fungus can be promising as a nematophagous fungus to control various species of root-knot nematodes.

It should be noted that a high nematicidal effect under the conditions of our laboratory and vegetative tests was shown by the biological standard—Nematofagin-Mikopro, the active substance of which is the fungus strain *A. oligospora* F-1303. Soliman M. S. et al. found high potency of *A. oligospora* culture filtrate at three different concentrations where immobilized/paralyzed or dead nematodes were observed, proving the nematicidal effect of *A. oligospora* metabolites. Also, the introduction of a suspension of the fungus into soil infected with *M. incognita* made it possible to reduce the formation of galls on tomato plants two times more than the control [40].

When assessing the nematicidal activity and biological efficacy of liquid cultures of microorganisms, strains of *Paecilomyces lilacinus* BK-6, *Metarhizium anisopliae* BK-2, and *Arthrobotrys conoides* BK-8 were isolated in the options where the highest mortality of J2 juveniles and a smaller number of formed galls on the roots of tomato plants were observed. In the future, we are going to determine their compatibility with each other and select the ratio of microorganisms to obtain effective mixtures, then create a multicomponent biological nematicide against meloidoginosis.

## 4. Materials and Methods

### 4.1. Research Location

Experimental studies were carried out at the Laboratory of Biorational Plant Protection Products and Technologies for Ecologized, Resource-saving, and Organic Agriculture of the Federal Research Center of Biological Plant Protection, Krasnodar, Russia.

### 4.2. Root-Knot Nematode Samples and Their Identification

Collection of the material with root-knot nematodes was carried out during phytosanitary monitoring of tomato plantations in the Rostov region (North-Western agro-climatic zone (47°34′20″ N 38°52′00″ E)), Krasnodar Krai (Central (45°02′00″ N 38°59′00″ E) and Anapo-Taman agro-climatic zones (45°16′00″ N 37°23′00″ E)), the Volgograd region (48°42′42″ N 44°30′50″ E), and the Republic of Dagestan (43°06′00″ N 46°53′00″ E).

To analyze the species composition of five *Meloidogyne* populations, PCR analysis was performed at the Center of Parasitology “A.N. Severtsov Institute of Ecology and Evolution of the Russian Academy of Sciences”, Moscow, Russia.

Root-knot nematodes in the amount of 1–5 individuals were placed in a test tube with a volume of 0.5 mL. Lysis was carried out using WLB+ solution: water, proteinase K (20 mg/mL); ribonuclease A solution (4 mg/mL); and 0.5M EDTA pH 8.0 solution. The prepared solution was added to a test tube with nematodes and incubated in a solid-state thermostat at 65 °C for 2 h. After the end of the isolation, the tubes with the isolated DNA were stored at −20 °C. Prior to PCR, the DNA samples were removed from the freezer and placed on a special chilled rack. Reagents for PCR were also placed on the rack.

PCR analysis was performed using the Encyclo Plus PCR Kit with two primer pairs (for the cytochrome oxidase 1 gene and the D2D3 28S rDNA region: D2A (ACA AGT ACC GTG AGG GAA AGT) and D3B (TGC GAA GGA ACC AGC TAC TA). PCR was performed according to the standard protocol: D2D3: primary DNA denaturation at 94 °C for 3 min followed by nine cycles consisting of denaturation at 94 °C for 1 min, annealing at 55 °C for 1.5 min and chain elongation at 72 °C for 1.5 min, followed by the next 24 cycles consisting of denaturation at 94 °C for 45 s, annealing at 57 °C for 1 min and chain elongation at 72 °C for 1 min 20 s. The reaction ended with a final elongation at 72 °C for 5 min Cytochrome oxidase 1: primary DNA denaturation at 94 °C for 3 min followed by 35 cycles consisting of denaturation at 94 °C for 45 s, annealing at 48 °C for 1 min and chain elongation at 72 °C 1 The reaction ended with a final elongation at 72 °C for 5 min, followed by cooling of the samples to a temperature of 22 °C. The results of the reaction were visualized in agarose gel.

The products of the PCR reaction were sent for Sanger sequencing [41] to the “Genome” Collective Use Centre (CUC “Genome”, Moscow, Russia). Subsequently, the obtained sequences were analyzed for the significance of differences from the sequences deposited in the Genbank using the MEGA 11 software package [42].

### 4.3. Obtaining a Suspension of Juveniles of the Northern Root-Knot Nematode (J2)

For research under the conditions of laboratory and vegetative tests, the most common type of root-knot nematode in the south of Russia was chosen. *M. hapla* was kept and propagated in vegetative pots on susceptible tomato plants. Egg sacs of the northern root-knot nematode (*M. hapla*) were taken from a parent culture (tomato). Under a binocular magnifier, the egg sacs were separated from the roots with dissecting needles, transferred to Petri dishes with water, and stored at room temperature (25 ± 3 °C). The volume of water was regularly aerated with a pipette. After 3 days, the juveniles that hatched from the eggs were collected with a pipette and put into a test tube for further studies [43].

### 4.4. Research Objects

In the experiment, we used liquid cultures of strains of root-knot nematode antagonist microorganisms provided from the working collection of BioTechAgro LLC (Krasnodar Krai, Russia) and a biological standard (Nematofagin-Mikopro) (Figure 4; Table 4).

The working collection of tomato marker forms collected by Academician A.A. Zhuchenko is being studied, maintained, and replenished in the Laboratory of Biorational Plant Protection Products and Technologies for Ecologized, Resource-saving, and Organic Agriculture of the Federal Research Center of Biological Plant Protection, created as part of the Scientific and Educational Center (SEC) of the South of Russia in 2021. The collection contains more than 300 genetically identified mutant lines. Employees carry out morphological description and culling of lines that do not correspond to the declared characteristics, as well as screening of tomato collection samples for resistance to major pests, including root-knot nematodes.

Under the conditions of the vegetative test, the mutant line Mo 463 was used, which is characterized by resistance to the tobacco mosaic virus (gene symbol: *Tm*-2^2^; chromosome, locus: 9 (22)) and susceptibility to root knot nematodes [44].

### 4.5. Effect of Liquid Cultures of Microorganism Strains on M. hapla

Under laboratory conditions, primary screening of microbial agents against second-stage juveniles (J2) of the northern root-knot nematode (*M. hapla*) was carried out. In 3 replications, 20 juvenile nematodes (J2) contained in 1 mL of suspension were placed in a sterile Petri dish with a diameter of 35 mm, then 1.5 mL of liquid cultures of microorganism strains were added. The number of dead nematodes was counted after 24 h, 48 h, and 72 h under a Bresser Researcher ICD LED 20×–80× stereoscopic microscope (manufacturer: Bresser, Rhede, Germany) [45]. Dead nematodes were considered immobilized, having taken a direct form. Sterile water with juvenile nematodes (J2) was taken as a control. The biological standard for comparison was Nematofagin-Mikopro (*Arthrobotrys oligospora* F-1303). The experiment was repeated three times.

### 4.6. Effect of Liquid Cultures of Microorganism Strains on the Presence of Galls on Tomato Plants

Determination of the nematicidal effect of liquid cultures of microorganism strains against the background of artificial soil infestation with *M. hapla* was carried out in vegetative pots on a susceptible mutant line (sample number: Mo 463) [43]. To prepare the working solution, 150 mL of water and 50 mL of each liquid culture (in a ratio of 3:1 for vegetable pots with a volume of 5 L) were used and applied to the soil infected with *M. hapla*. After 4 days of treatment with liquid cultures, 50-day-old tomato seedlings of a mutant line susceptible to the northern root-knot nematode were planted in the soil of all options. The control was the option without treatment.

Biological efficacy against *M. hapla* was determined 180 days after tomato planting by the number of galls per plant according to the formula of Abbott, 1925 [46,47]:BE = 100 × (1 − A/B),(1)
where BE is the biological efficiency or percentage of mortality of individuals, %; A is the number of individuals in the experimental option after treatment; and B is the number of individuals in the control.

### 4.7. Statistical Analysis

Statistical data processing was carried out by standard methods using MS Excel and the ANOVA program for Windows. All data were expressed as the mean of three replicates ± standard deviation (SD). Duncan’s test was used, and differences were considered statistically significant at a *p* < 0.05 level.

## Figures and Tables

**Figure 1 plants-12-03323-f001:**
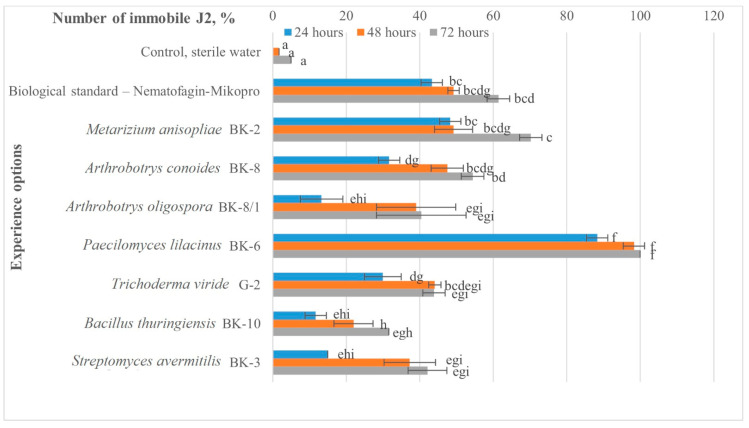
The number of immobile second-stage juveniles (J2) of *M. hapla* in the experimental options with the application of microbial strains, %. Bars show the mean *n* = 3 repetitions of each treatment ± standard error. The columns marked with the same letters in each experimental option do not differ significantly, according to the test for the least significant difference (*p* ≤ 0.05).

**Figure 2 plants-12-03323-f002:**
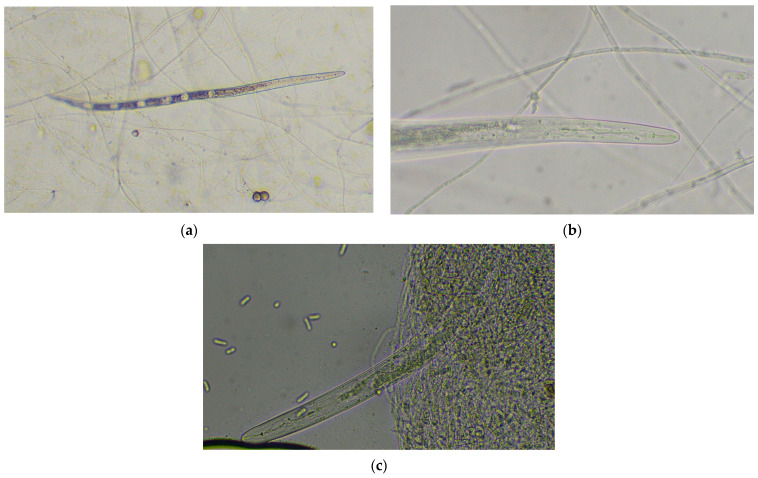
Immobilized *M. hapla* juvenile exposed to strains of antagonist microorganisms after 48 h: (**a**) *P. lilacinus* BK-6 at 150× magnification; (**b**) *M. anisopliae* BK-2 at 600× magnification; (**c**) Nematofagin-Mikopro at 600× magnification.

**Figure 3 plants-12-03323-f003:**
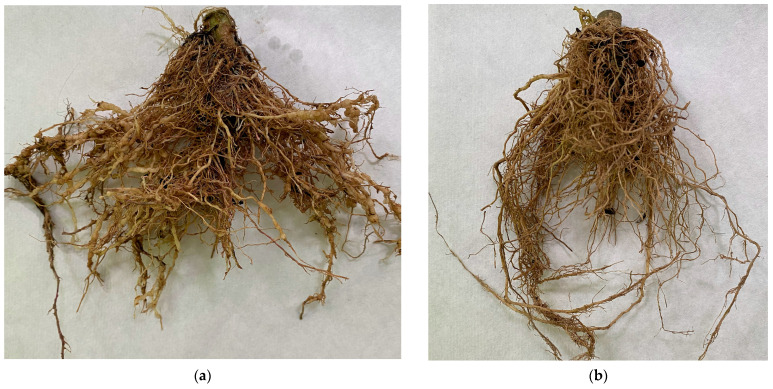
Root of susceptible mutant line (Mo 463) infected by northern root-knot nematode: (**a**) control (infected plants, water); (**b**) option with *P. lilacinus* strain BK-6.

**Figure 4 plants-12-03323-f004:**
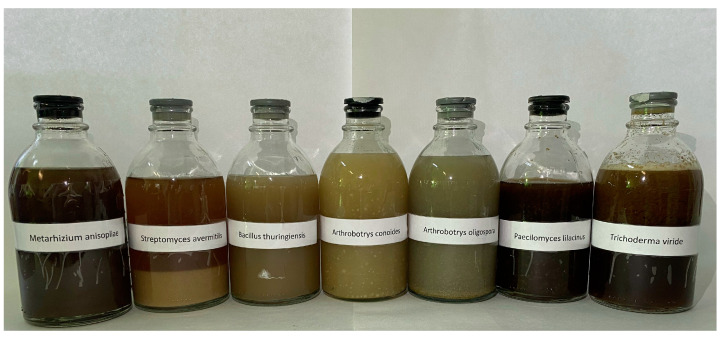
Liquid cultures of antagonist microorganisms provided by BioTechAgro LLC for in vitro and in vivo screening.

**Table 1 plants-12-03323-t001:** Genetic distance matrix built in the MEGA11 program based on the cox1 gene sequences of the studied samples of the northern root-knot nematode *Meloidogyne hapla*, as well as sequences from the NCBI Genbank.

	M_hapla_1	M_hapla_2	M_hapla_3	M_hapla_4	M_hapla _MH128580	M_exigua _MH128478	M_incognita _MH128456	M_javanica_ MH128449	M_arenaria _MH128444	M_floridensis _MH128431
M_hapla_1										
M_hapla_2	0.00137931									
M_hapla_3	0.00137931	0.00000000								
M_hapla_4	0.00275862	0.00137931	0.00137931							
M_hapla_MH128580	0.00690608	0.00828729	0.00828729	0.00966851						
M_exigua_MH128478	0.14364641	0.14502762	0.14502761	0.14502762	0.13674033					
M_incognita_MH128456	0.14583333	0.14722222	0.14722222	0.14861111	0.13888889	0.14305556				
M_javanica_MH128449	0.14722222	0.14861111	0.14861111	0.15000000	0.14027778	0.14444444	0.00277778			
M_arenaria_MH128444	0.14722222	0.14861111	0.14861111	0.15000000	0.14027778	0.14444444	0.00277778	0.00000000		
M_floridensis_MH128431	0.14722222	0.14861111	0.14861111	0.15000000	0.14027778	0.14444444	0.00277778	0.00000000	0.00000000	

**Table 2 plants-12-03323-t002:** Genetic distance matrix built in the MEGA11 program based on the cox1 gene sequences of the studied samples of the southern root-knot nematode *Meloidogyne incognita*, as well as sequences from the NCBI Genbank.

	M_incognita_1	M_incognita _MH743219	M_hapla _MH128580	M_luci _MG969511	M_javanica _OR038715	M_inornata _ku372168	M_ethiopica _ku372162	M_paranaensis _OQ750577	M_hispanica _OQ842292
M_incognita_1									
M_incognita_MH743219	0.00000000								
M_hapla_MH128580	0.12271541	0.12271541							
M_luci_MG969511	0.01190476	0.01190476	0.13315927						
M_javanica_OR038715	0.01666667	0.01666667	0.13054830	0.02857143					
M_inornata_KU372168	0.02619048	0.02619048	0.13577024	0.03809524	0.03809524				
M_ethiopica_KU372162	0.02857143	0.02857143	0.14621410	0.03095238	0.04047619	0.05000000			
M_paranaensis_OQ750577	0.02676399	0.02676399	0.15223097	0.03406326	0.04379562	0.04866180	0.05352798		
M_hispanica_OQ842292	0.02985075	0.02985075	0.1436031	0.04228856	0.04228856	0.04228856	0.05472637	0.05750000	

**Table 3 plants-12-03323-t003:** Effect of liquid cultures of microorganisms on the presence of galls on roots of tomato plants (Mo 463), FSBSI FRCBPP, 2022.

Experimental Option	Quantity of the Preparation, mL	Number of Studied Plants	Number of Infected Plants	Number of Galls/Plant (*n* = 40)	Biological Efficiency, %
Control (infected plants, water)	-	40	40	97.6 ± 6.1	-
Control (uninfected plants, water)	-	0	0	0.0	-
Biological standard—Nematofagin-Mikopro	50	40	19	17.7 ± 1.9	82.0
*Metarisium anisopliae* BK-2	50	40	21	23.9 ± 2.3	75.7
*Arthrobotrys conoides* BK-8	50	40	28	25.2 ± 1.7	74.5
*Arthrobotrys oligospora* BK-8/1	50	40	31	55.7 ± 3.6	42.9
*Paecilomyces lilacinus* BK -6	50	40	19	18.5 ± 2.4	81.2
*Trichoderma viride* G-2	50	40	35	47.2 ± 3.3	51.3
*Bacillus thuringiensis* BK-10	50	40	38	61.9 ± 2.8	36.8
*Streptomyces avermitilis* BK-3	50	40	35	58.3 ± 2.3	40.4

**Table 4 plants-12-03323-t004:** Root-knot nematode antagonist microorganism strains used in the research.

Type of Microorganism	Strain	Titer, Colony-forming Units (CFU)/mL
*Metarisium anisopliae*	BK-2	(7.0 ± 0.5) × 10^7^
*Arthrobotrys conoides*	BK-8	(6.0 ± 0.3) × 10^6^
*Arthrobotrys oligospora*	BK-8/1	(5.5 ± 0.1) × 10^7^
*Paecilomyces lilacinus*	BK-6	(3.2 ± 0.8) × 10^7^
*Trichoderma viride*	G-2	(5.5 ± 0.2) × 10^6^
*Bacillus thuringiensis*	BK-10	(5.0 ± 0.2) × 10^9^
*Streptomyces avermitilis*	BK-3	(3.0 ± 0.1) × 10^7^
*Arthrobotrys oligospora* (Nematofagin-Mikopro)	F-1303	(3.0 ± 0.2) × 10^6^

## Data Availability

Data are contained within the article.

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
