# Peer review of "Primary Screening of Microorganisms against Meloidogyne hapla (Chitwood, 1949) under the Conditions of Laboratory and Vegetative Tests on Tomato"

_plants, 2023, doi:10.3390/plants12183323_

Round 1

Reviewer 1 Report

Dear Aurthors,

it is an interesting article that needs minor changes before pubblishing. Please find here after my comments:

Line 31: “meloidogynosis disease” not sure of the term!

At places the format of scientific names needs to be changed in italics (e.g. 2nd paragraph of introduction, Figure 1 and 2, line 170  etc).

Thorough language editing.  

Please include in the article title the species composition study of root-knot nematodes of Russia.

Line 112: Please explain “vegetation vessels.”

Lune 161: Please change “their mortality” to “nematode juveniles mortality”.

Line 349: Please explain in what context did you choose  the mutant line Mo 463 for the experiments

Line 357: Please explain in what bases did you chose as a test concentration 2.5 ml of liquid culture per Petri dish

Line 370: Please express the test concentration as volume of liquid culture per gr of soil.

Line 380: “Scope of work” is confusing here! Explain or delete

Please explain “experimental option” in Figure 3. Is it assessment time?

To my opinion authors should stress more the originality of their work compared to what has been done till now on the topic. In specific, e.g the originality of the microbial strains in Russia to treat against RKN ..

Please see to a moderate language and format (italics etc) editing throughout the whole text.

Author Response

The authors worked on the article and took into account the comments on the text.
The changes made in the text are marked in yellow.

  1. Corrected the name of our institution (line 6). Therefore, the lines got lost further along the text.
    Lines 61-64 were combined into one paragraph (became: lines 60-63). Combined the paragraph on line 234 with line 235.
  2. On line 17 (renumbered from line 18) changed from «Primary screening» to «Screening».
  3. Lines 61-64 were combined into one paragraph (became: lines 60-63). Combined the paragraph on line 234 with line 235.
  4. On page 4, the name was corrected from «Figure 1», «Figure 2» to «Table 1», «Table
  5. The numbering of the literature in the text was changed after [30].
  6. The authors have made adjustments to the lines 390-392:

Funding: The research was carried out in accordance with the State Assignment of the Ministry of Science and Higher Education of the Russian Federation within the framework of research on the topic No FGRN‐2021‐0001.   Responses to the reviewer's comments:  

Question: Line 31: “meloidogynosis disease” not sure of the term!
      Answer:  Root-knot (gall) nematodes (Meloidogyne spp.; RKNs) are highly adapted obligate endoparasites of the root system that cause a disease known as meloidoginosis. Some articles by Russian scientists, scientists from Brazil and other scientists indicate the name of this disease (for example: https://doi.org/10.1016/j.plaphy.2011.10.008 , https://doi.org/10.31016/978-5-6046256-9-9.2022.23.296-301 ).  

Question: At places the format of scientific names needs to be changed in italics (e.g. 2nd paragraph of introduction, Figure 1 and 2, line 170  etc).

      Answer:  Fixed. The Latin of nematodes and fungi is highlighted in italics in the text. Highlighted in yellow.  

Question: Thorough language editing. 

      Answer:  Fixed.  

Question: Please include in the article title the species composition study of root-knot nematodes of Russia.
      Answer: The title of the article was renamed to «Species Diversity of Nematodes of the Genus Meloidogyne on Tomato in the Conditions of Southern Russia and Screening of Microorganisms for a Common Species».  

Question: Line 112: Please explain “vegetation vessels.”

      Answer: Fixed on «vegetation pots» (Line 110). Highlighted in yellow.  

Question: Lune 161: Please change “their mortality” to “nematode juveniles mortality”.

      Answer: Fixed (Line 158-159). Highlighted in yellow.  

Question: Line 349: Please explain in what context did you choose  the mutant line Mo 463 for the experiments

      Answer: Fixed on «Under the conditions of the vegetative test, the mutant line Mo 463 was used, which is characterized by resistance to the tobacco mosaic virus (gene symbol: Tm-22; chromosome, locus: 9 (22)) and susceptibility to root knot nematodes» (line 346).  

Question: Line 357: Please explain in what bases did you chose as a test concentration 2.5 ml of liquid culture per Petri dish

      Answer: We made a mistake and corrected it by 1.5 ml. So many liquid cultures of strains of microorganisms are needed for a Petri dish with a diameter of 35 mm. (Line  352)  

Question: Line 370: Please express the test concentration as volume of liquid culture per gr of soil.

      Answer: «in a ratio of 3:1 for vegetable pots with a volume of 5 liters» (line 364-365).  

Question: Line 380: “Scope of work” is confusing here! Explain or delete

      Answer: Deleted “Scope of work”  

Question: Please explain “experimental option” in Figure 3. Is it assessment time?

      Answer: The experimental options in this figure are the strains of microorganisms used, the control and the biological standard. Fixed “experimental option” on “experimental options”.  

The authors express their gratitude to the reviewers for their comments on the article.

Reviewer 2 Report

In this MS, a series of experiments were carried out for primary screening of microorganisms against Meloidogyne hapla under the conditions of laboratory and vegetative tests on tomato. The topic is very interesting and good design. While there are some shortcomings and errors, which were following as:

 Q1: In this MS, the tested microorganisms were antagonist microorganisms provided by “Biotechagro” LLC, not just microorganisms. So it is not primary screening of microorganisms against M. hapla, it should be screening of antagonist microorganisms against M. hapla.

 Q2: Introduction and Discussion: Too many short paragraphs in these sections. Revise and combine some short paragraphs.

 Q3: Results: L113-116: Remove this paragraph to the M&M section.

 Q4: Figure 1 and Figure 2: These two figures should be tables not figures.

 Q5: Figure 3: The letters marked on data columns are confused to show bcdg and dg (where e?),  bd(where c?), ehi (where f, g?), egi where f, h?etc.

Author Response

The authors worked on the article and took into account the comments on the text.
The changes made in the text are marked in yellow.

Corrected the name of our institution (line 6). Therefore, the lines got lost further along the text.

 Responses to the reviewer's comments:
Point 1. Q1: In this MS, the tested microorganisms were antagonist microorganisms provided by “Biotechagro” LLC, not just microorganisms. So it is not primary screening of microorganisms against M. hapla, it should be screening of antagonist microorganisms against M. hapla.

Answer:  The title of the article was renamed to «Species Diversity of Nematodes of the Genus Meloidogyne on Tomato in the Conditions of Southern Russia and Screening of Microorganisms for a Common Species»

On line 17 (renumbered from line 18) changed from «Primary screening» to «Screening».

 Point 2. Q2: Introduction and Discussion: Too many short paragraphs in these sections. Revise and combine some short paragraphs.

Answer:  Lines 61-64 were combined into one paragraph (became: lines 60-63). Combined the paragraph on line 234 with line 235.

 Point 3. Q3: Results: L113-116: Remove this paragraph to the M&M section.

Answer:  Paragraph «Selected populations of root-knot nematodes were used for DNA isolation followed by Sanger sequencing of the cytochrome oxidase subunit 1 (cox1) region. Subsequently, the obtained sequences were analyzed for the significance of differences with the sequences deposited in the Genbank using the MEGA 11 software package [31].». We decided to remove it because this information is repeated in the materials and methods : «The products of the PCR reaction were sent for Sanger sequencing [41] to the “Genome” Collective Use Centre (CUC “Genome”, Moscow). Subsequently, the obtained sequences were analyzed for the significance of differences with the sequences deposited in the Genbank using the MEGA 11 software package [42].»

 Point 4. Q4: Figure 1 and Figure 2: These two figures should be tables not figures.

Answer:  On page 4, the name was corrected from «Figure 1», «Figure 2» to «Table 1», «Table

 Point 5. Q5: Figure 3: The letters marked on data columns are confused to show bcdg and dg (where e?),  bd(where c?), ehi (where f, g?), egi (where f, h?)etc.

Answer: As indicated in the caption of Figure 1 (former figure 3, renumbered) «The columns marked with the same letters in each experimental option do not differ significantly, according to the test for the least significant difference». Accordingly, if the letters in the variants are not repeated, then the difference between them is significant. And if there is the same letter in the variants, then there is no difference between them.

The authors have made adjustments to the lines 390-392:

 Funding: The research was carried out in accordance with the State Assignment of the Ministry of Science and Higher Education of the Russian Federation within the framework of research on the topic No FGRN‐2021‐0001.

The numbering of the literature in the text was changed after [30].

The authors express their gratitude to the reviewers for their comments on the article.
